# Natural Sunlight Driven Photocatalytic Removal of Toxic Textile Dyes in Water Using B-Doped ZnO/TiO$_2$ Nanocomposites

Romana Akter Shathy [1], Shahriar Atik Fahim [1], Mithun Sarker [1], Md. Saiful Quddus [2], Mohammad Moniruzzaman [3], Shah Md. Masum [1,*] and Md. Ashraful Islam Molla [1,*]

[1] Department of Applied Chemistry and Chemical Engineering, Faculty of Engineering and Technology, University of Dhaka, Dhaka 1000, Bangladesh; s-2013912615@acce.du.ac.bd (R.A.S.); s-2015216823@acce.du.ac.bd (S.A.F.); mithun@du.ac.bd (M.S.)

[2] Institute of Glass and Ceramic Research and Testing, Bangladesh Council of Scientific and Industrial Research, Dhaka 1205, Bangladesh; mdsaifulquddus@gmail.com

[3] Central Analytical and Research Facilities, Bangladesh Council of Scientific and Industrial Research, Dhaka 1205, Bangladesh; monirbcsir@gmail.com

* Correspondence: masumacce@du.ac.bd (S.M.M.); ashraful.acce@du.ac.bd (M.A.I.M.); Tel.: +88-017-4946-7276 (S.M.M.); +88-015-5235-9706 (M.A.I.M.)

**Abstract:** A novel B-doped ZnO/TiO$_2$ (B–ZnO/TiO$_2$) nanocomposite photocatalyst was prepared using a mechanochemical–calcination method. For the characterization of the synthesized B–ZnO/TiO$_2$ photocatalyst, XRD, FESEM-EDS, FTIR, UV-Vis DRS, BET, PL, and XPS techniques were used. The bandgap energy of B–ZnO/TiO$_2$ was reduced, resulting in enhanced visible-light absorption. Significant PL quenching confirmed the reduction in the electron–hole recombination rate. Furthermore, reduced crystallite size and a larger surface area were obtained. Hence, the B–ZnO/TiO$_2$ photocatalyst exhibited better photocatalytic activity than commercial TiO$_2$, ZnO, B–ZnO, and ZnO/TiO$_2$ in the removal of methylene blue (MB) dye under natural sunlight irradiation. The effects of various parameters, such as initial concentration, photocatalyst amount, solution pH, and irradiation time, were studied. Under optimal conditions (MB concentration of 15 mg/L, pH 11, B–ZnO/TiO$_2$ amount of 30 mg, and 15 min of operation), a maximum MB removal efficiency of ~95% was obtained. A plausible photocatalytic degradation mechanism of MB with B–ZnO/TiO$_2$ was estimated from the scavenger test, and it was observed that the $\bullet O_2^-$ and $\bullet OH$ radicals were potential active species for the MB degradation. Cyclic experiments indicated the high stability and reusability of B–ZnO/TiO$_2$, which confirmed that it can be an economical and environmentally friendly photocatalyst.

**Keywords:** B–ZnO/TiO$_2$; photocatalyst; nanocomposite; mechanochemical; methylene blue; sunlight

## 1. Introduction

Synthetic dyes are employed extensively in a variety of industries, including textiles, leather, cosmetics, and paint industries. Each year, over 0.8 million tons of different types of dyes are produced worldwide [1]. Approximately 15% of the world's total dye production is discarded and discharged in textile effluents [1,2]. The transfer of these colored waste streams into the ecosystem without proper treatment harms the environment and marine life and poses serious health threats to humans as they are toxic, recalcitrant, mutagenic, and carcinogenic [1]. For instance, methylene blue (MB, a thiazine cationic dye) has adverse health effects, which include breathing difficulties, vomiting, eye burns, diarrhea, and nausea when MB is accumulated in wastewater. In addition, due to its nonbiodegradability, it is highly persistent in the environment [1]. There are different conventional methods for the treatment of textile dyes in water, such as physical methods, biological methods, chemical precipitation, membrane filtration, reverse osmosis, ozonation, filtration, adsorption, ultrasonic-assisted adsorption, incineration, and coagulation [1,3–5].

Each treatment method, however, has certain drawbacks, including high energy waste, high cost, and the generation of secondary pollutants. Traditional physical methods are nondestructive, but instead of eradicating pollutants, they only transfer pollutants to other media, generating secondary pollution [4,6]. Even though biological methods are commonly used commercially, they still require the wastewater effluent to be diluted, as microalgae are susceptible to concentrated toxicity [3].

Advanced oxidation processes (AOP), particularly heterogeneous photocatalysis, have recently emerged as a potential destructive method for the total mineralization of most organic contaminants. Photocatalysis is a method that uses light as an energy source to activate a catalyst that speeds up chemical reactions without actually being consumed in the reaction. The photocatalytic reaction has several advantages, including the ability to occur at room temperature as well as almost complete mineralization of organic pollutants into $CO_2$, $H_2O$, and $N_2$. As a semiconductor photocatalyst, metal oxides such as ZnO, $TiO_2$, $ZrO_2$, $SnO_2$, and $WO_3$ are employed [7]. Among these, ZnO and $TiO_2$ show similar properties. These include similar bandgap energy, better electro-optical properties, strong oxidation ability, high electron mobility, low toxicity, and an environmentally friendly nature [8,9]. In spite of having many favorable characteristics, there are still some drawbacks to ZnO and $TiO_2$. Among the shortcomings, the most challenging ones for $TiO_2$ photocatalyst to overcome are (1) the large bandgap (ZnO = 3.37 eV and $TiO_2$ = 3.2 eV), which requires ultraviolet illumination [5,10], and (2) the high electron–hole pair recombination rate [5,7]. Additionally, in the presence of UV light, ZnO experiences photocorrosion problems as well as photoinstability and low quantum yield in aqueous solutions. Moreover, the most active phase of $TiO_2$, i.e., anatase, is thermally unstable [7]. To overcome these shortcomings, a variety of techniques have been employed on ZnO and $TiO_2$. These include metal and nonmetal doping, noble metal loading, dye sensitization, and semiconductor coupling [11–16]. Panwar et al. [17] fabricated Gd–ZnO/$TiO_2$ nanocomposites via the sol–gel method and studied the photocatalytic degradation of organic dyes under UV irradiation. Kerli et al. [18] employed a hydrothermal method to synthesize Ag–ZnO/$TiO_2$ nanocomposite particles for photocatalytic dye degradation. Li et al. [19] synthesized ZnO/$TiO_2$–B composite photocatalyst through a sol–gel method and evaluated the photodegradation of 4-chlorophenol under visible light irradiation. Wang et al. [20] used a microwave hydrothermal technique to produce C–ZnO/$TiO_2$ composites and investigated the photocatalytic degradation efficiency of RhB under simulated sunlight irradiation.

The use of semiconductor coupling to enhance photocatalytic activity is a promising approach. Semiconductor coupling has been discovered to facilitate charge separation and inhibit electron–hole recombination [21,22]. This also provides additional benefits of increased charge transfer to adsorbed substrates, enhanced electro-optical properties, enhanced transport properties, and increased charge carrier period [7,19]. Moreover, the ionic radii of $Zn^{2+}$ (0.74) and $Ti^{4+}$ (0.75) are very close, which offers great potential for improving photocatalytic activity by semiconductor coupling of ZnO and $TiO_2$. Furthermore, nonmetals such as B, C, F, N, S, and others are now being employed to reduce electron–hole recombination, enhance grain size, and increase surface area [23]. Because of the atomic size and electronic structure of B, B doping has attracted much attention [24]. In addition, the introduction of B inhibits the crystal growth, allowing a photocatalyst with a higher surface area to be achieved [9]. In view of these advantages, B and ZnO/$TiO_2$ could be combined to yield extraordinary nanocomposites to achieve better photocatalytic activity. However, to the best of our knowledge, reports on the synthesis of B–ZnO/$TiO_2$ nanocomposites using a simple mechanochemical–calcination method are scarce.

In this study, we synthesized B–ZnO/$TiO_2$ nanocomposites and applied them for the removal of methylene blue (MB) dye from water. The effects of several parameters, such as photocatalyst amount, initial dye concentration, pH, and irradiation time, were investigated. Finally, the photocatalytic mechanism of the prepared nanocomposite photocatalyst was suggested on the basis of the radical-quenching experiments. This study's B–ZnO/$TiO_2$ nanocomposites showed the rapid removal of MB (~95% within 15 min) from water, which

is considered a major breakthrough. Such a fast removal certainly strengthens the industrial application of the material for toxic dye removal from industrial effluents. The stability of the B–ZnO/TiO$_2$ was also confirmed for use in environmental purification.

## 2. Results and Discussion

### 2.1. XRD Study

This XRD analysis of commercial TiO$_2$, ZnO, B–ZnO, ZnO/TiO$_2$, and B–ZnO/TiO$_2$ nanoparticles is shown in Figure 1. According to Figure 1, there are diffraction peaks at $2\theta = 25.3°, 36.9°, 37.8°, 38.6°, 48.1°, 53.9°, 55.1°, 62.1°, 62.7°$, and $68.8°$ with corresponding planes (101), (103), (004), (112), (200), (105), (211), (213), (204), and (116), respectively, which are associated with anatase phases of TiO$_2$ (JSPDS No. 21–1272) [25]. The distinct peaks obtained for ZnO at $2\theta = 31.8°, 34.4°, 36.3°, 47.5°, 56.6°, 62.9°, 66.4°, 68°$ and $69.1°$ correspond to the planes (100), (002), (101), (102), (110), (103), (200), (112), and (201), respectively (JSPDS No. 36–1451) [23,26]. The Scherrer equation, $D = 0.9\lambda/\beta cos\theta$, was used to calculate the crystallite sizes of TiO$_2$, ZnO, B–ZnO, ZnO/TiO$_2$, and B–ZnO/TiO$_2$ from the full width at half maximum (FWHM) of the corresponding crystal peaks, as shown in Table 1. Here, D is the average crystallite size, $\lambda$ is the X-ray wavelength, $\theta$ is the Bragg diffraction angle, and $\beta$ is the full width at half maximum. Additionally, peaks observed for B–ZnO/TiO$_2$ were found to have both peaks associated with ZnO and TiO$_2$ combined, and the intensity for selective peaks was decreased compared with pristine TiO$_2$ and ZnO, which indicates a smaller crystallite size formed, providing enhanced photodegradation efficiency [23].

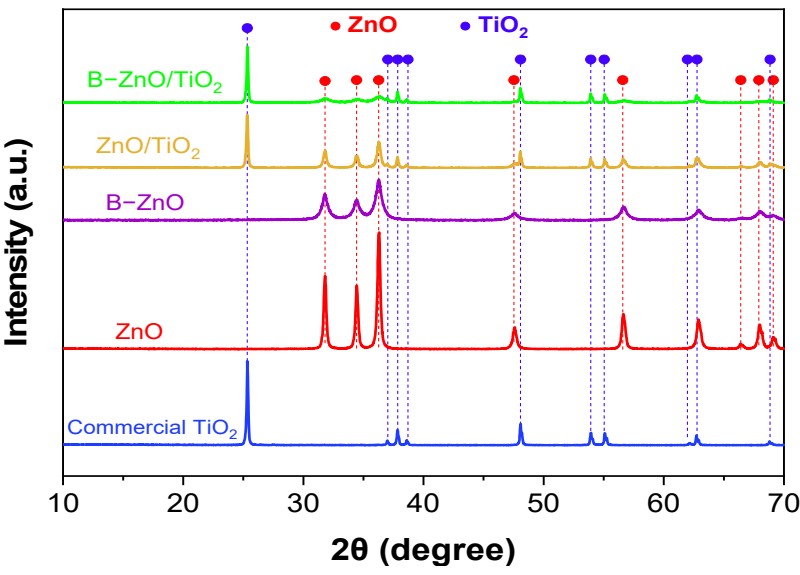

**Figure 1.** XRD diffraction patterns of TiO$_2$, ZnO, B−ZnO, ZnO/TiO$_2$, and B−ZnO/TiO$_2$.

**Table 1.** Crystallite size, BET parameters, and bandgap energies.

| Photocatalysts | Crystallite Size (nm) | BET Parameters | | Bandgap Energies (eV) | |
|---|---|---|---|---|---|
| | | BET Surface Area (m$^2$/g) | Pore Volume (cm$^3$/g) | Direct Transitions | Indirect Transitions |
| Commercial TiO$_2$ | 73.35 | 10.76 | 0.0371 | − | 3.21 |
| ZnO | 32.10 | 7.30 | 0.0732 | 3.22 | − |
| B–ZnO | 14.03 | − | − | 3.18 | − |
| ZnO/TiO$_2$ | 52.40 | − | − | 2.93 | 3.13 |
| B–ZnO/TiO$_2$ | 42.54 | 18.99 | 0.0781 | 2.89 | 3.06 |

## 2.2. FESEM Study

The surface, size, and particle morphologies of the commercial $TiO_2$, synthesized ZnO, B–ZnO, $ZnO/TiO_2$, and B–$ZnO/TiO_2$ photocatalysts evaluated by FESEM are visualized in Figure 2. In Figure 2a, $TiO_2$ shows globular-shaped fine nanoparticles with a homogeneous distribution, as described in the literature [27]. The micrograph of ZnO reveals its morphological appearance is polycrystalline in nature. Most of the particles possess smooth surfaces and are polygonal in shape, which indicates that the samples were polycrystalline wurtzite structure, as presented in Figure 2b [28]. Images of B–ZnO are of irregular spheroid structure structures with a greater extent of agglomeration (Figure 2c). Particle size was also reduced after the incorporation of boron (B) into ZnO, which indicates successful doping [23]. As shown in Figure 2d, by incorporating ZnO into $TiO_2$, the observed morphology seems to retain properties of both ZnO and $TiO_2$ with a higher degree of agglomeration. The micrograph of B–$ZnO/TiO_2$ demonstrates smaller spherical-shaped particles with less agglomeration compared to both ZnO and $TiO_2$ (Figure 2e,f). Generally, B doping deteriorates the crystallinity of ZnO to a certain extent [29], because the radius of $B^{3+}$ (0.02 nm) is smaller than that of $O^{2-}$ (0.14 nm) and $Zn^{2+}$ (0.074 nm). As a result, when the O or Zn atoms are replaced with B, the crystal plane spacing shrinks, causing the diffraction peaks to shift to a greater angle. Consequently, the B atoms are likely to occupy the octahedral interstices [29]. However, Figure 2e exhibited substantially larger particles because the FESEM image of the heterogeneous semiconductor photocatalyst B–$ZnO/TiO_2$ was focused where $TiO_2$ nanoparticles were more predominant. To make it easier to understand, another FESEM picture of B–$ZnO/TiO_2$ nanocomposites has been presented (Figure 2f), which shows characteristics of both B–ZnO and $TiO_2$.

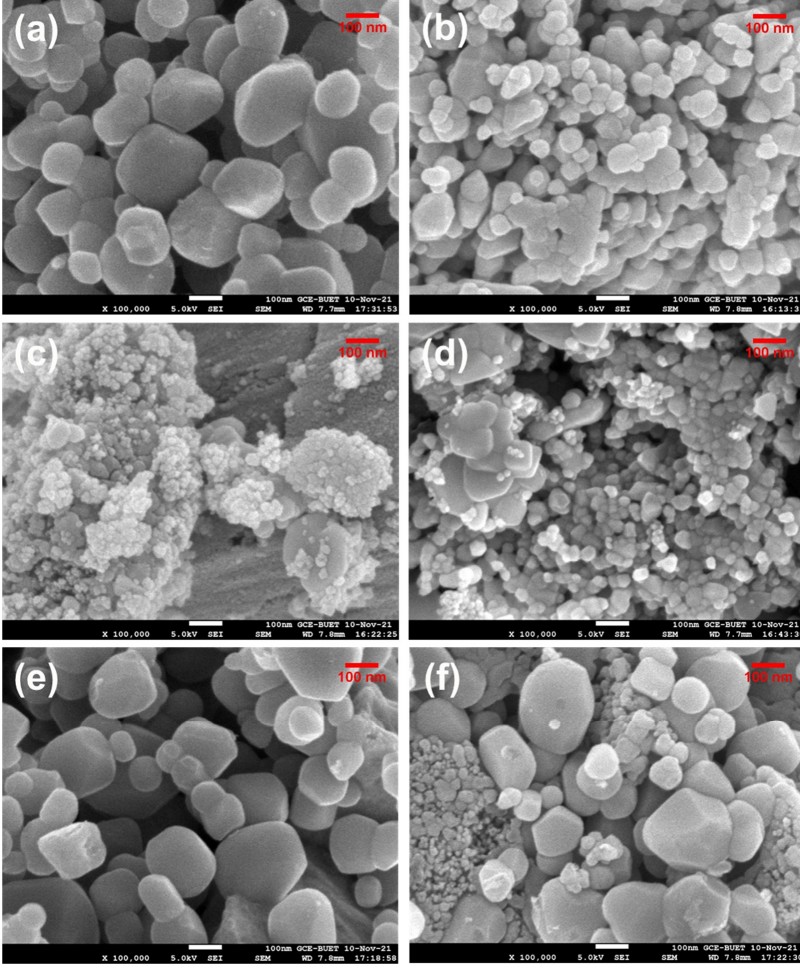

**Figure 2.** FESEM images of (**a**) $TiO_2$, (**b**) ZnO, (**c**) B–ZnO, (**d**) $ZnO/TiO_2$, and (**e**,**f**) B–$ZnO/TiO_2$.

The calculations for average particle size of nanocomposites were carried out through ImageJ software. The nanoparticles were marked at 59 different locations for $TiO_2$, 132 for ZnO, and 61 for $B–ZnO/TiO_2$ using Figure 2a, Figure 2b, and Figure 2e, respectively, and plotted as a size distribution histogram (Figure S1a–c). The average particle size of nanocomposites is calculated to be 114, 53, and 88 nm for $TiO_2$, ZnO, and $B–ZnO/TiO_2$ respectively. The average particle size measured by FESEM shows a good agreement with the XRD.

### 2.3. SEM and EDS Mapping Study

SEM and EDS mapping examinations were carried out on synthesized $B–ZnO/TiO_2$ nanocomposite to determine the presence of boron (B) atom in the sample (Figure S2). From Figure S2b–e, it was observed that the various elements of B, Zn, Ti, and O were distributed throughout the sample, confirming the presence of all elements and the successful synthesis of $B–ZnO/TiO_2$ nanocomposites.

### 2.4. EDS Study

Figure S3 show the energy-dispersive X-ray spectroscopy (EDS) line-scanning outputs of commercial $TiO_2$, ZnO, B–ZnO, $ZnO/TiO_2$, and $B–ZnO/TiO_2$ photocatalysts. From the EDS spectrum of commercial $TiO_2$ (Figure S3a), sharp peaks of elemental Ti and O have been found. Figure S3b shows the strong peaks of elemental Zn and O in the EDS spectra of ZnO. Peaks related to elements B, Zn, and O are found in the EDS spectra of B–ZnO (Figure S3c). Figure (S3d) depicts EDS spectra of $ZnO/TiO_2$ with sharp peaks for the elements Zn, Ti, and O. From the EDS spectra analysis of $B–ZnO/TiO_2$ shown in Figure S3e, peaks related to elements of B, Zn, Ti, and O have been found, confirming the presence of all four elements. No other peaks of different elements were found, which confirms the formation of a pure $B–ZnO/TiO_2$ composite. The elemental mass percentage and atom percentage of all photocatalysts are presented in Table S1. From Table S1, the obtained values for mass percentage and atom percentage are very close to stoichiometric values. However, composites containing boron, namely B–ZnO and $B–ZnO/TiO_2$, show an exception. The amount of boron present in the photocatalysts could not be measured even though a peak of boron exists in the EDS spectra. This was due to the inability of the EDS instrument to detect elements with lower molecular weight [30]. However, when the mass percentage of B–ZnO was compared to ZnO and $B–ZnO/TiO_2$ was compared to $ZnO/TiO_2$, it was discovered that the mass percentage changed significantly with the increased amount of oxygen present in boron-doped composites. This ensures changes in composition with the incorporation of boron in B–ZnO and $B–ZnO/TiO_2$.

### 2.5. FTIR Study

Figure 3 illustrates the Fourier transform infrared spectroscopy (FTIR) results of commercial $TiO_2$, ZnO, B–ZnO, $ZnO/TiO_2$, and $B–ZnO/TiO_2$ photocatalysts. The FTIR spectrum of $TiO_2$ displays a broad band between 470 and 865 $cm^{-1}$ owing to the overlapping of several bands assigned to Ti–O–Ti and Ti–O vibration modes [31]. The intense peak near 675 $cm^{-1}$ is attributed to the Ti–O stretching [32]. The major characteristic peaks obtained for ZnO between 460 and 540 $cm^{-1}$ are because of the absorption band with the stretching mode of Zn–O [23,26]. The peaks observed between 1600 and 700 $cm^{-1}$ are responsible for the B–O stretching vibrations. This confirms the successful doping of boron into ZnO. The IR spectrum assigned to $ZnO/TiO_2$ illustrates the fingerprint region of both ZnO and $TiO_2$. Furthermore, $B–ZnO/TiO_2$ introduces a weak peak at 1300 $cm^{-1}$ associated with B–O stretching vibrations. The stretching and bending vibrations of H–O–H and O–H showed signals in the ranges of 3000–3800 $cm^{-1}$ and 1600–1650 $cm^{-1}$. From the intensity of these peaks, it can be affirmed that the moisture content of all photocatalysts is negligible. No other peaks are found in the IR spectra. Therefore, each photocatalyst is free of any type of contamination, which is consistent with the EDS results.

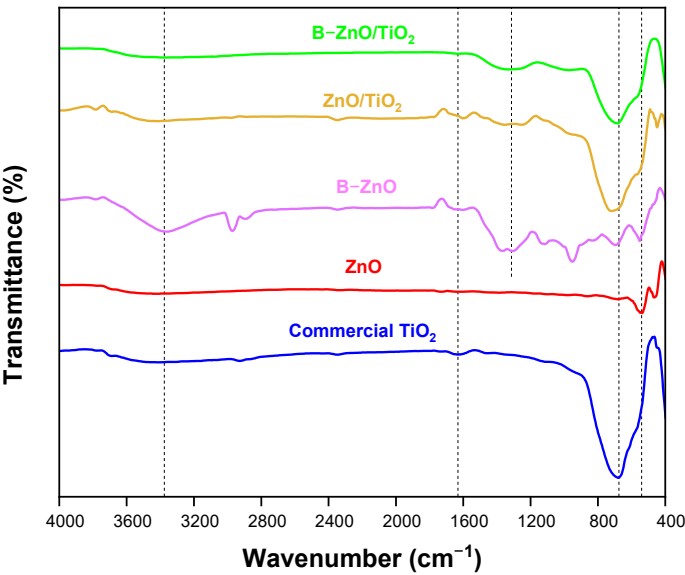

**Figure 3.** FTIR spectra of $TiO_2$, ZnO, B−ZnO, $ZnO/TiO_2$, and B−ZnO/$TiO_2$.

### 2.6. UV-Vis DRS Study

Figure 4a represents the diffuse reflectance spectra (DRS) measured from 300 nm to 700 nm. From the DRS, commercial $TiO_2$, ZnO, B−ZnO, $ZnO/TiO_2$, and B−ZnO/$TiO_2$ photocatalysts show optical absorption below 400 nm. The photocatalysts demonstrate high reflectance at visible wavelengths. A rapid rise in reflectance has been observed for $TiO_2$, ZnO, B−ZnO, $ZnO/TiO_2$, and B−ZnO/$TiO_2$ at the absorption edge at 414, 440, 430, 455, and 411 nm, respectively.

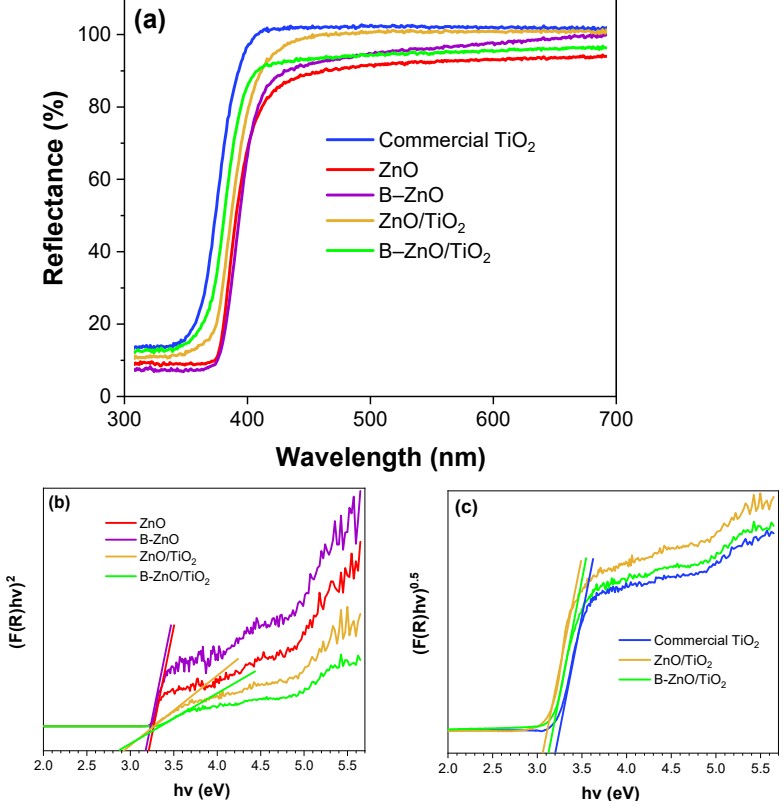

**Figure 4.** (**a**) UV-Vis DRS patterns of $TiO_2$, ZnO, B−ZnO, $ZnO/TiO_2$, and B−ZnO/$TiO_2$; Tauc plot for (**b**) direct and (**c**) indirect transitions.

The Kubelka–Munk function given by Equation (1) has been used to calculate the bandgap $E_g$, where equivalent absorption coefficient is obtained from reflectance $R$.

$$F(R) = (1 - R)^2 / 2R \qquad (1)$$

where F($R$) represents the Kubelka–Munk function and is proportional to the absorption coefficient $\alpha$, and $R$ is the reflectance. By extrapolating the straight portion of the plot of $(\alpha h\nu)1/n$ versus $h\nu$ to the $h\nu$ axis of the Tauc equation, the bandgap of the photocatalyst can be determined.

$$F(R) \times h\nu = A\,(h\nu - E_g)^n \qquad (2)$$

Here, $A$ is the proportionality constant and $h$ is Planck's constant ($6.63 \times 10^{-34}$ Js$^{-1}$). In this case, for direct allowed transitions, n = 1/2, and for indirect transitions, n = 2. Studies revealed that the direct bandgap is followed by ZnO and TiO$_2$ follows the indirect bandgap. So, both n = 0.5 and n = 2 were applied to the B$-$ZnO/TiO$_2$ photocatalysts to obtain a more accurate bandgap (Figure 4b,c) [33]. The estimated bandgap energies of photocatalysts TiO$_2$, ZnO, B$-$ZnO, ZnO/TiO$_2$, and B$-$ZnO/TiO$_2$ are presented in Table 1. The B$-$ZnO/TiO$_2$ photocatalyst exhibits a decrease in bandgap energy compared with TiO$_2$ and ZnO owing to the introduction of B$-$ZnO with TiO$_2$ [34]. The lowest values of bandgap are obtained for B$-$ZnO/TiO$_2$ (~2.89 eV for n = 0.5 and ~3.06 eV for n = 2). The electrons from the conduction band injected from ZnO to TiO$_2$ may be responsible for the bandgap narrowing. This is also favorable to electron–hole (e$^-$/h$^+$) separation and increased photocatalytic activity of B–ZnO/TiO$_2$. Moreover, the bandgap energies of B$-$ZnO/TiO$_2$ for both direct and indirect transitions are consistent with previously published literature [20].

*2.7. BET Study*

Figure 5 demonstrates N$_2$ adsorption/desorption isotherms of commercial TiO$_2$, ZnO, and B$-$ZnO/TiO$_2$ photocatalysts. According to the IUPAC classification, the adsorption/desorption isotherms are found to be of type IV. Table 1 summarizes the BET surface area of TiO$_2$, ZnO, and B$-$ZnO/TiO$_2$ photocatalysts. The BET surface areas of ZnO and TiO$_2$ are 7.30 and 10.76 m$^2$/g, respectively, whereas B$-$ZnO/TiO$_2$ has a surface area of 18.99 m$^2$/g, which is consistent with previous findings [35]. Interestingly, B$-$ZnO/TiO$_2$ exhibits a higher BET surface area than bare ZnO and TiO$_2$. The increase in surface area of B$-$ZnO/TiO$_2$ may be associated with the incorporation of boron, which prevents agglomeration of the ZnO and TiO$_2$ in the nanocomposite, and is also in agreement with FESEM analysis. As a result, the addition of boron increases the surface area and also increases the number of active sites. This suggests that the synergistic influence of adsorption and photocatalysis of B$-$ZnO/TiO$_2$ enables the removal of more MB dye molecules from water [36].

*2.8. PL Study*

Figure 6 presents the photoluminescence (PL) spectra of commercial TiO$_2$, ZnO, B$-$ZnO, ZnO/TiO$_2$, and B$-$ZnO/TiO$_2$ photocatalysts. The photocatalysts were all excited at 325 nm at room temperature. The PL spectrum of TiO$_2$ displays one significant peak at 376 nm, whereas ZnO shows a broad hump at 425–600 nm in the visible region. The broad hump near 553 nm corresponds to the oxygen vacancies, whereas the peak at 382.5 nm refers to the band edge emission [37,38]. The PL spectrum of B$-$ZnO/TiO$_2$ photocatalyst reveals a small band edge emission at 380 nm and a very weak peak in the visible region. Hence, the decay in the PL intensity suggests a decline in the recombination rate of photogenerated e$^-$/h$^+$ pairs of B$-$ZnO/TiO$_2$. Moreover, the PL intensity is weakened for B$-$ZnO compared with bare ZnO, suggesting an efficient electron capture by the doped B energy level [19]. Thus, combining B$-$ZnO with the TiO$_2$ nanoparticles enhanced the separation efficiency of e$^-$/h$^+$ in B$-$ZnO/TiO$_2$ which consequently improved the photocatalytic activity [39]. Moreover, the crystal quality can be evaluated using UV emission

luminescence, whilst the occurrence of structural defects can be detected using visible light emission. Relatively low photoluminescence intensity suggests a slow rate of electron–hole recombination, while a very large PL intensity indicates a fast rate of recombination. During the fast reaction formation process, surface and subsurface defects occurred, resulting in higher intensity. Furthermore, highly crystalline materials exhibit a higher PL peak intensity than amorphous materials [21].

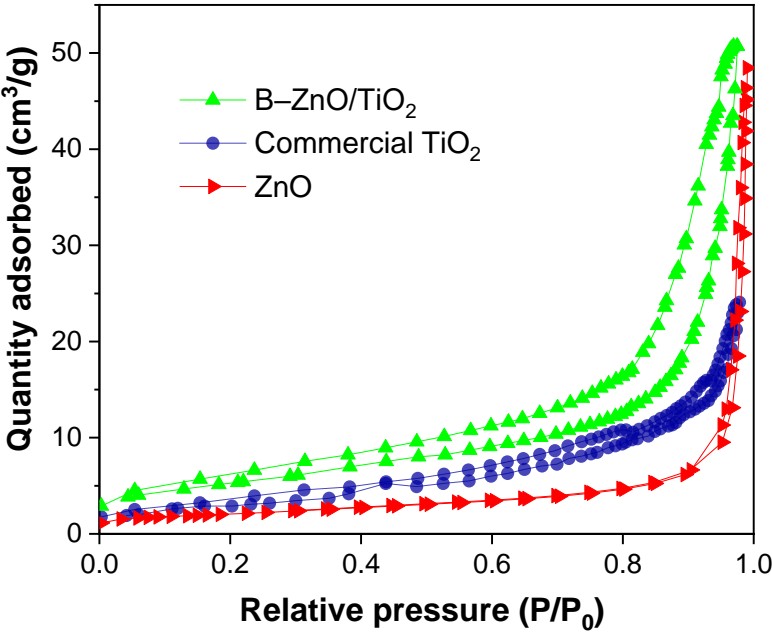

**Figure 5.** Nitrogen adsorption–desorption isotherms of $TiO_2$, ZnO, and $B–ZnO/TiO_2$.

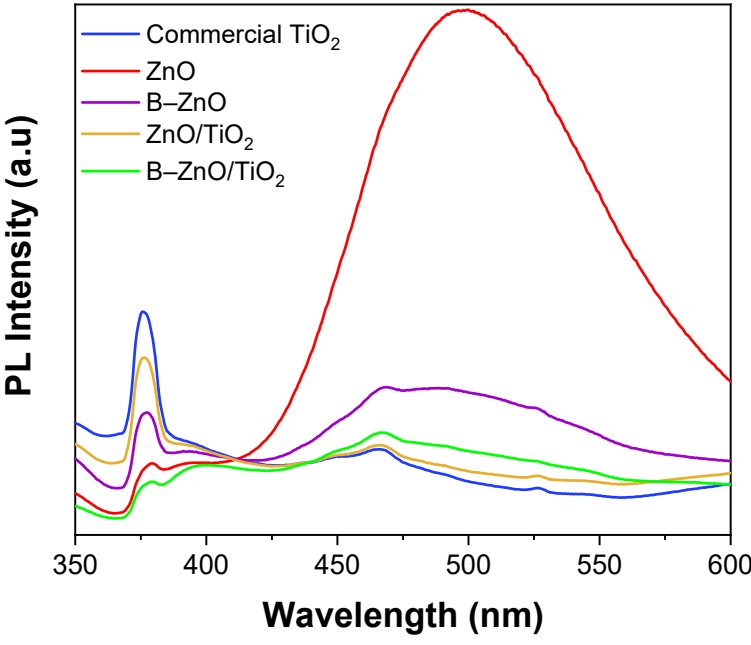

**Figure 6.** PL spectra of $TiO_2$, ZnO, B–ZnO, $ZnO/TiO_2$, and $B–ZnO/TiO_2$.

## 2.9. XPS Study

X-ray photoelectron spectroscopy (XPS) was used to analyze the surface compositions and related valence states of the $B−ZnO/TiO_2$ photocatalyst. Figure 7a shows the XPS full survey spectrum, which clearly shows the peaks of Zn, Ti, B, O, and C elements. During

the synthesis process, the C ingredient could have come from hydrocarbons. As a result, the nanocomposites are only made up of Zn, Ti, B, and O, and these findings are consistent with the XRD patterns. The high-resolution spectra of Zn 2p are shown in Figure 7b. The Zn $2p_{3/2}$ and Zn $2p_{1/2}$ of $Zn^{2+}$ are responsible for the peaks centered at 1022.2 and 1045.3 eV, respectively [23]. Figure 7c displays two peaks in the Ti 2p XPS spectrum, Ti $2p_{3/2}$ (458.6 eV) and Ti $2p_{1/2}$ (464.4 eV) [9,20]. Figure 7d shows the binding energy peak of B1s at 192.3 eV, showing that the doping B atoms are in the trivalent state of $B^{3+}$ [9,23]. Furthermore, the O1s XPS spectra of B−ZnO/TiO$_2$ (Figure 7e) reveal that the peak at 529.9 eV is attributable to the lattice oxygen anions ($O^{2-}$) [23,35]. The peak at 531.2 eV relates to weakly bound oxygen species such C−O and C=O groups, whereas the peak at 532.2 eV indicates the presence of OH groups on the nanocomposite surface [20,23].

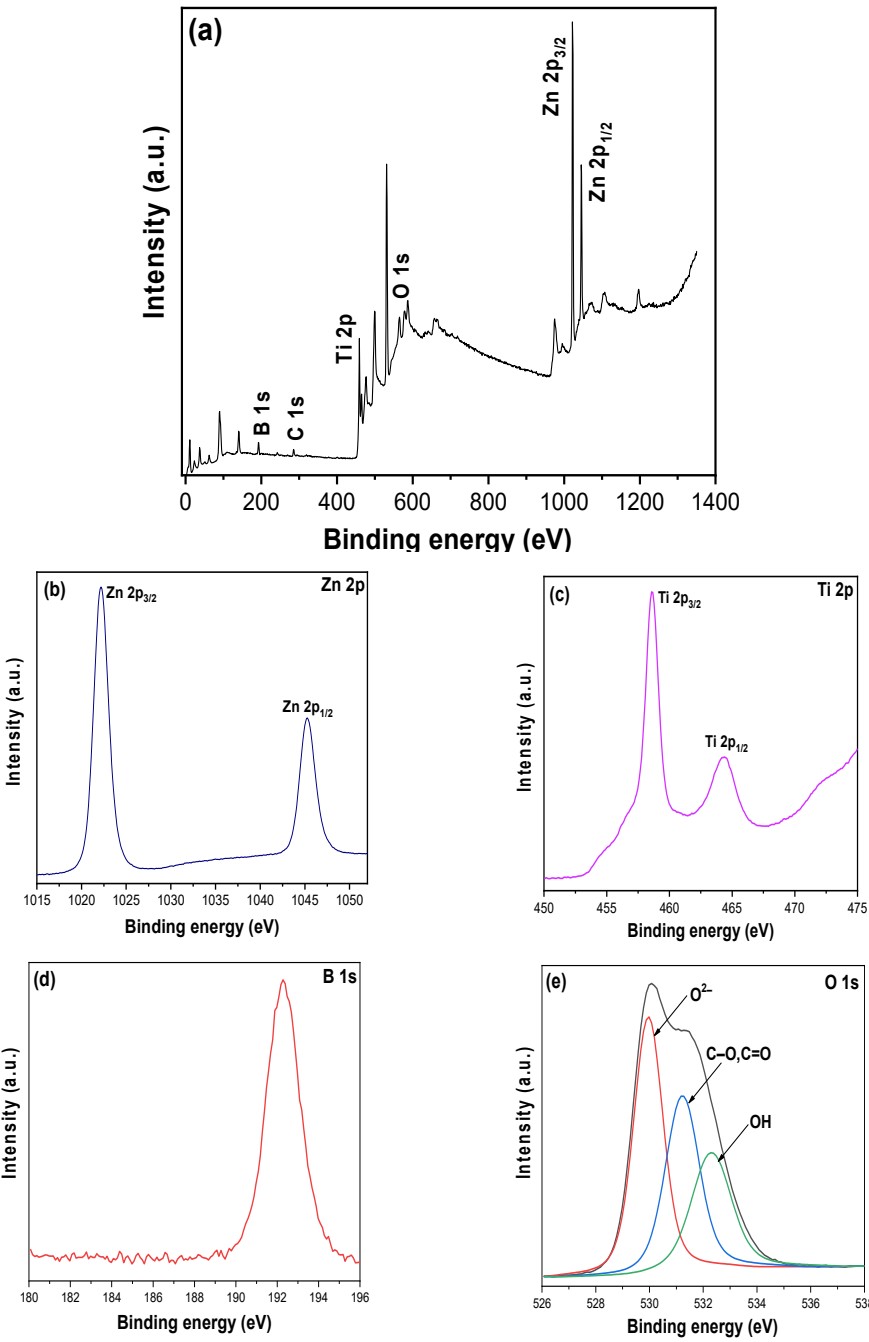

**Figure 7.** XPS spectra of B–ZnO/TiO$_2$ (**a**) survey, (**b**) Zn 2p, (**c**) Ti 2p, (**d**) B 1s, and (**e**) O 1s.

### 2.10. Photocatalytic Removal of MB Dye

The UV–visible absorption spectra of MB dye with commercial $TiO_2$, ZnO, B–ZnO, $ZnO/TiO_2$, and B–$ZnO/TiO_2$ photocatalysts are illustrated in Figure 8a (natural pH 6) and Figure 8b (optimized pH 11). The intensity of the major peak of MB ($\lambda_{max}$ = 662 nm [11]) decreases with various photocatalysts. The peak intensity decreases with reduced dye concentration due to the photocatalytic removal in the presence of photocatalysts and sunlight irradiation. Photolysis of MB at a natural pH of 6 and an optimized pH of 11 was performed for 15 min under sunlight without a photocatalyst to monitor self-degradation (Figure 8c). It is observed that the self-degradation of MB dye solution is negligible at both pH levels during the photocatalytic test [40]. Without sunlight irradiation, another set of experiments was conducted to determine the removal efficiency of MB in the dark because of the dye adsorption on the B–$ZnO/TiO_2$ photocatalyst surface. In the dark, the removal efficiency of MB with B–$ZnO/TiO_2$ is 10.52% and 48.58% at pH 6 and pH 11, respectively. At basic pH, B–$ZnO/TiO_2$ showed greater dye removal efficiency. This can be explained as, at basic conditions, the positively charged cationic MB dye is easily absorbed on the strongly negatively charged photocatalyst surface.

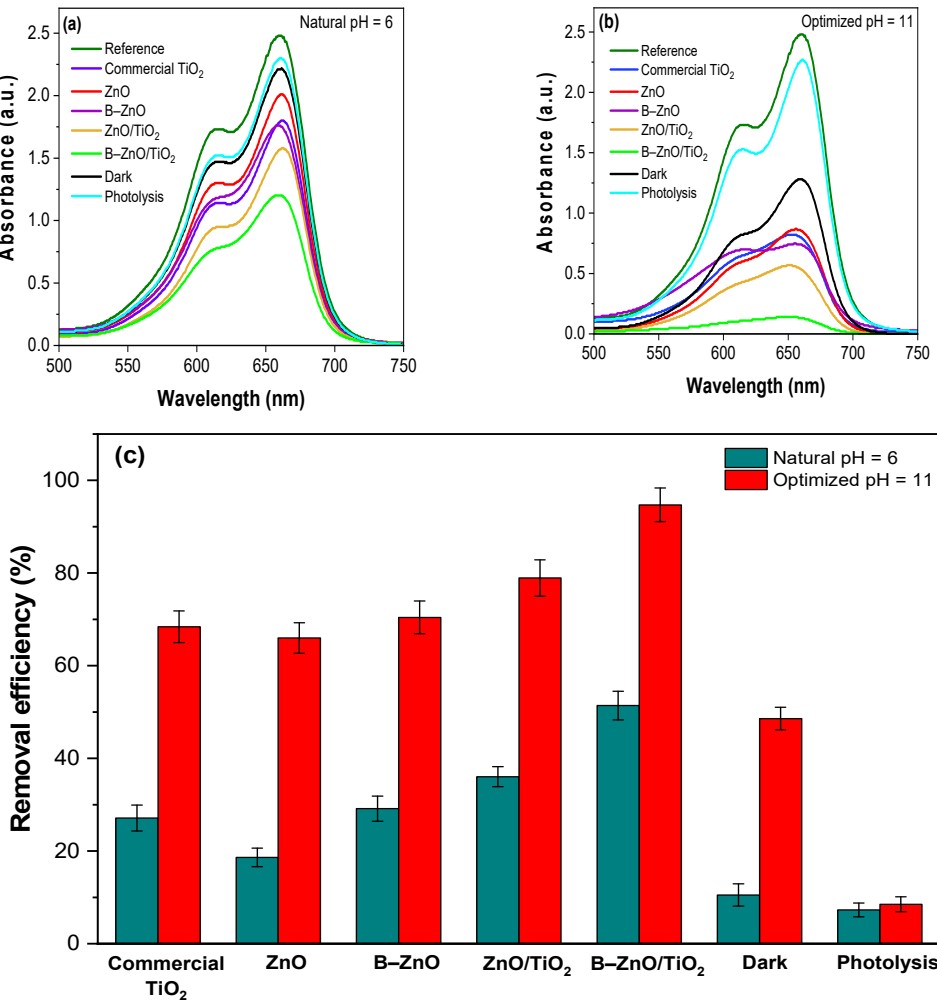

**Figure 8.** (**a**,**b**) UV–visible results showing the decay in the intensity of MB solution; (**c**) Photocatalytic removal, adsorption under dark, and photolysis for MB (initial MB concentration: 15 mg/L; photocatalyst amount: 30 mg; pH: 6 and 11; irradiation time: 15 min).

The understanding of the photocatalytic removal efficiency of synthesized photocatalysts is of great importance. A series of experiments were conducted with commercial $TiO_2$, ZnO, B–ZnO, $ZnO/TiO_2$, and B–$ZnO/TiO_2$ photocatalysts at natural pH 6 and opti-

mized pH 11 while keeping other parameters constant, as shown in Figure 8c. From the photocatalytic experiments, the MB removal efficiencies of commercial $TiO_2$, ZnO, B–ZnO, $ZnO/TiO_2$, and $B–ZnO/TiO_2$ were found to be 27.12%, 18.62%, 29.14%, 36.03%, and 51.41%, respectively, at the natural pH of dye (pH 6). The MB removal efficiencies at optimized pH 11 were found to be 68.42%, 65.99%, 70.44%, 78.94%, and 94.73%, respectively, after 15 min of irradiation. There is a significant rise in the MB removal efficiency of $B–ZnO/TiO_2$ compared with pristine $TiO_2$ and ZnO. It also exceeds B–ZnO and $ZnO/TiO_2$ by a sufficient amount. This can be explained as the coupling of $TiO_2$ with ZnO suppresses the $e^-/h^+$ recombination rate [40,41], which is consistent with the PL results. Moreover, boron doping can substantially inhibit crystal size growth [9,19], which is confirmed by XRD, hence increasing photocatalytic activity [9]. Furthermore, the surface area of $B–ZnO/TiO_2$ increases and the bandgap energy decreases for $B–ZnO/TiO_2$, as confirmed by BET and UV-Vis DRS data, respectively. Thus, B, ZnO, and $TiO_2$ altogether resulted in a synergistic effect to obtain enhanced photocatalytic performance of $B–ZnO/TiO_2$ [17,26]. Furthermore, $B–ZnO/TiO_2$ nanocomposites exhibit superior photocatalytic performance compared to other nanocomposites presented in the literature [17,18,20,40–45], as summarized in Table 2, indicating that $B–ZnO/TiO_2$ can be a promising photocatalyst for the removal of organic dyes.

**Table 2.** Comparison of the photocatalytic activity of $B–ZnO/TiO_2$ nanocomposites with recently reported doped $ZnO/TiO_2$ nanocomposites for organic dye removal.

| Nanocomposites | Synthesis Methodology | Dye Concentration | Photocatalyst Dimension | Time (min) | Light Source | Removal Efficiency | Ref. |
|---|---|---|---|---|---|---|---|
| $B–ZnO/TiO_2$ | Mechanochemical–calcination method | MB (15 mg/L) | 30 mg/50 mL | 15 | Natural sunlight (~1.2 mW/cm$^2$) | ~95% | Present study |
| $Gd–ZnO/TiO_2$ | Sol–gel method | MB (20 mg/L) MO (20 mg/L) | 1 g/L | 90 | Fluorescent lamps (30 W) | 93% (MB) 94% (MO) | [17] |
| $Ag–ZnO/TiO_2$ | Hydrothermal method | MB, RhB, MG (5 mg/L) | 10 mg/50 mL | 120 | Xenon lamp (300 W) | 99% (MB) 87% (RhB) 71% (MG) | [18] |
| $C–ZnO/TiO_2$ | Hydrothermal–calcination method | RhB (10 mg/L) | 50 mg/100 mL | 45 | Xenon lamp (200 W) | 94% | [20] |
| $ZnO/TiO_2–CNFs$ | Hydrothermal method | MB (10 mg/L) | 20 mg/25 mL | 120 | Ultraviolet lamp ($\lambda$ = 365 nm) | 93% | [40] |
| $Cu–TiO_2/ZnO$ | Sol–gel method | MO MB | 0.7 g/L | 120 | Fluorescent lamp (18–23 W) | 85% (MO) 73% (MB) | [41] |
| $S–ZnO/TiO_2$ | Sol–gel method | RhB ($2.5 \times 10^{-5}$ mol/L) | 50 mg/100 mL | 100 | Halogen lamps (125 W) | 92% | [42] |
| $Ni–ZnO/TiO_2$ | Sol–gel method | RBB (50 mg/L) | 1 g/L | 120 | Simulated sunlight (350 W Xe lamp) | 61% | [43] |
| $Au–ZnO/TiO_2$ | Hydrothermal | MO (10 mg/L) | 3.34 mg/50mL | 300 | Mercury lamp (300 W) | 95% | [44] |
| $ZnO/TiO_2–rGO$ | Solvothermal method | MB (20 mg/L) | 252.5 mg/L | 63.5 | Ultraviolet lamp (11 W) | 99% | [45] |

## 2.11. Effect of pH

The pH of the reaction medium has a substantial impact on the photocatalytic performance of the catalyst. This is because the surface characteristics and the size of aggregated nanocomposites are influenced by pH. The pH can also influence the charge of dye molecules as well as the concentration of reactive hydroxyl radicals [5]. To understand the pH effect, experiments were conducted with the pH varying from 3 to 12 while the other parameters remained constant. Figure 9 represents the experimental outcomes. Overall, MB removal efficiency increases with increasing pH, and the maximum stands at the most basic pH 12, more than 98.32% degradation within 15 min. A rapid increasing trend is found from pH 3 to pH 10, with a removal efficiency of between 20.59% and 88.69%, and then it rises steadily. At pH 11, which is taken as the optimum pH, MB removal efficiency of 94.73% was obtained. Because textile waste effluent has a characteristic pH of 10 [46], pH 11 will be feasible and effective for almost complete removal of dye without introducing any additional pollution. The pH effect can be explained on the basis of electrostatic attraction between the photocatalyst and the dye. The photocatalyst's surface charge properties are influenced by the acidity or basicity of the solution, which has a direct impact on its photoactivity. The photocatalyst becomes less attracted to the dye molecules at zero charge point ($pH_{PZC}$). The literature reveals that $pH_{PZC}$ is ~6.0 for $TiO_2$ [40,47] and ~9.0 for $ZnO$ [40,48]. Therefore, the protonation reaction occurs at pH < $pH_{PZC}$, and the surface of the catalyst becomes positively charged. In contrast, the deprotonation reaction occurs at pH > $pH_{PZC}$, resulting in a negatively charged catalyst surface [5,40]. MB is a cationic dye with a pH of ~6 in solution [5]. At acidic pH, the positively charged B–$ZnO/TiO_2$ photocatalyst and the cationic MB dye are electrostatically repulsed, resulting in reduced photoactivity. On the other hand, at basic conditions, a strong electrostatic attraction facilitates the positively charged cationic dye MB to be adsorbed on the strongly negatively charged B–$ZnO/TiO_2$ surface. This electrostatic interaction facilitates the adsorptive property, which consequently enhances the photocatalytic removal efficiencies. These findings are completely consistent with previously published reports [40].

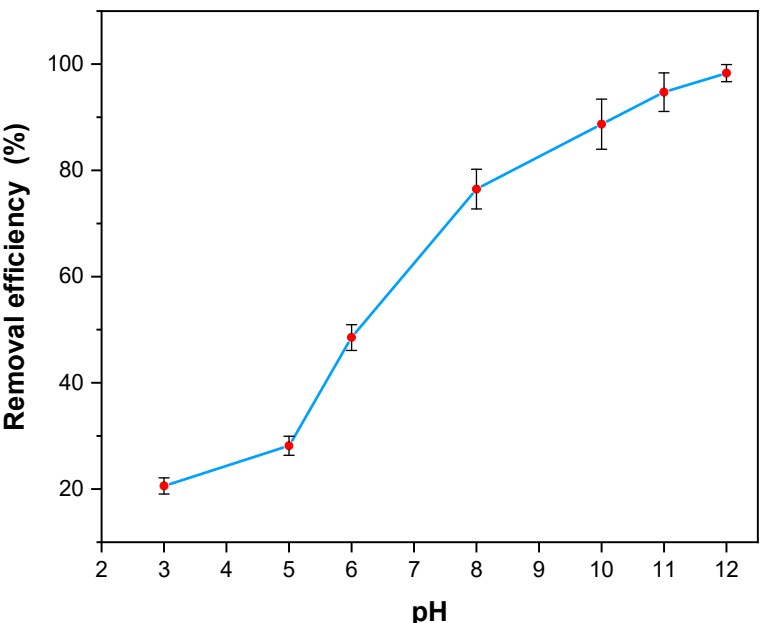

**Figure 9.** Effect of pH on the photocatalytic removal of MB with B–$ZnO/TiO_2$ (initial MB concentration: 15 mg/L; photocatalyst amount: 30 mg; pH: 3–12; irradiation time: 15 min).

## 2.12. Effect of Photocatalyst Dosages

The photocatalytic removal efficiency has a direct relation to the amount of photocatalyst used during the photodegradation process. This is due to the fact that the efficiency

of photocatalytic removal is dependent on the photon absorption capacity of the catalysts and the active site availability of the catalysts [9]. To investigate the effects of photocatalyst dosage on the photocatalytic removal of MB dye, experiments were conducted with the most active photocatalyst B–ZnO/TiO$_2$ by varying the amount, ranging from 10 to 50 mg (Figure 10). It is noticed that the MB removal efficiency increased linearly from 63.13% to 94.73% with the increase in photocatalyst loading from 10 to 30 mg. However, the removal efficiency declined to 83.13% and 73.19% on further loading of the photocatalyst up to 40 and 50 mg, respectively. This suggests that the MB degradation has a maximum dosage of 30 mg. The increase in removal efficiency with increasing catalyst loading can be explained as the availability of active sites in the photocatalyst is higher compared to the MB molecules present. At low photocatalyst dosages, the number of catalyst active sites is insufficient for higher amounts of dye molecules. As a result, only a limited number of active radicals are produced. With increased photocatalyst, the generation of active radicals, for example, superoxide and hydroxyl radicals, increases, resulting in more free electrons in the conduction band and higher light absorption. Therefore, the photocatalytic activity is significantly enhanced [7,9]. When the photocatalyst loading is very high, the catalyst may scatter light photons, and its turbidity may completely block UV light penetration. As a result, the photocatalytic removal efficiency is reduced. Another factor that contributes to low efficiency is the agglomeration of catalyst particles due to excess catalyst. As a result, the exposed surface area of the catalyst is reduced, and the catalyst becomes inactive [7,49].

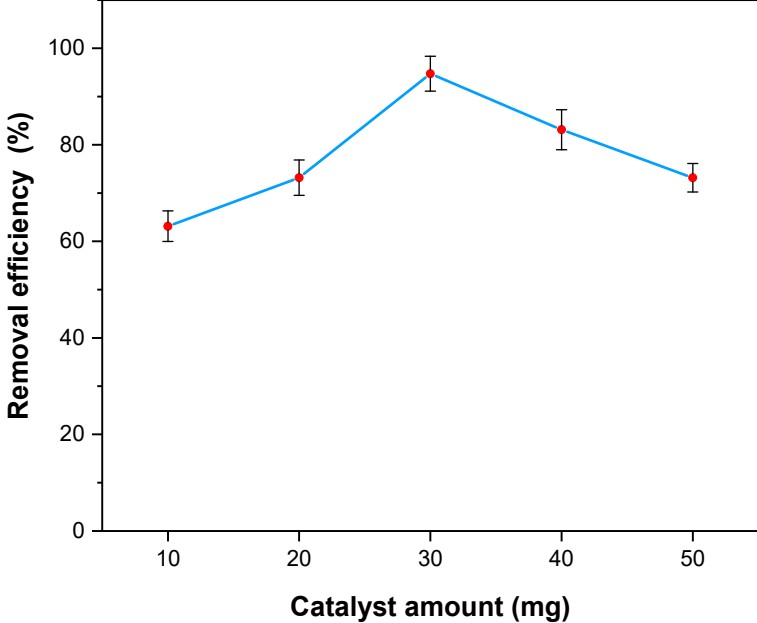

**Figure 10.** Effect of photocatalyst amount on the photocatalytic removal of MB with B–ZnO/TiO$_2$ (initial MB concentration: 15 mg/L; photocatalyst amount: 10–50 mg; pH: 11; irradiation time: 15 min).

*2.13. Effect of Initial MB Dye Concentration*

The initial MB dye solution concentrations varying from 5 to 25 mg/L were used at an optimum photocatalyst amount of 30 mg and pH of 11 to study the effect on the removal efficiency (Figure 11). When the concentration of MB gradually increased from 5 to 25 mg/L, the removal efficiency gradually declined from 98.75% to 70.46%. Additionally, the amount of dye removed increases as the initial dye concentration increases. A steeper slope can be seen between 5 and 15 mg/L, and the slope then gradually flattens off up to 25 mg/L. This phenomenon can be explained as occurring when, at higher dye concentrations, the number of active radicals present in a sample becomes insufficient for the degradation of a higher number of dye molecules. Additionally, when the number of

molecules increases, light penetration is impeded, and prior to reaching the catalyst active surface, photons are interrupted, reducing photocatalytic removal efficiency. Moreover, as the initial concentration of MB is increased, more intermediates are produced, which become adsorbed on the catalyst's active surface. This interrupts the formation of active radicals and decreases the number of active sites available in the photocatalyst [7,11,40]. Due to the concentration of real textile wastewater, 15 mg/L of MB solution was used for the following experiment to investigate the dye removal efficiency.

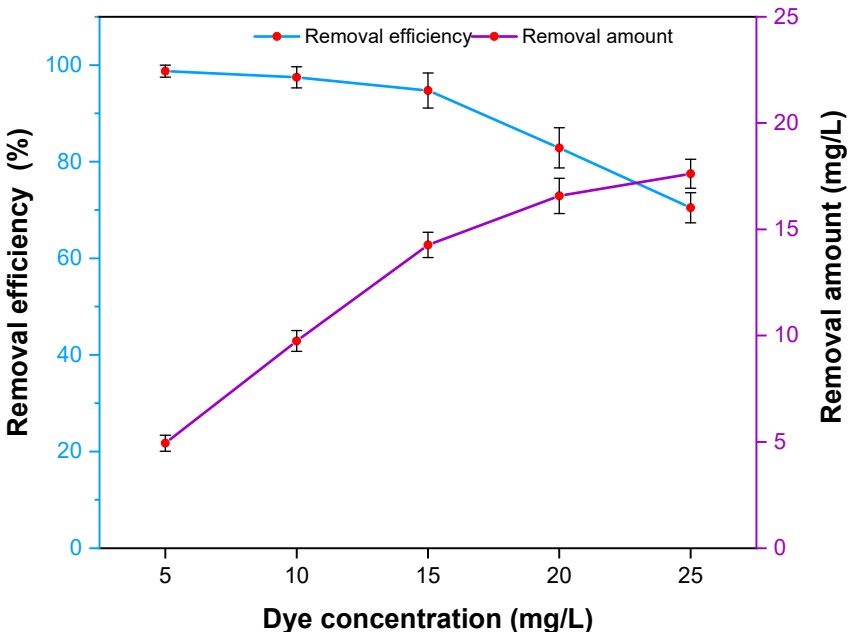

**Figure 11.** Effect of initial dye concentration on the photocatalytic removal of MB with B–ZnO/TiO$_2$ (initial MB concentration: 5–25 mg/L; photocatalyst amount: 30 mg; pH: 11; irradiation time: 15 min).

### 2.14. Effect of Irradiation Time

Figure 12 shows MB removal efficiency using B–ZnO/TiO$_2$ photocatalyst at different irradiation times (5–25 min) under sunlight irradiation. MB removal efficiency increased when increasing the irradiation time. Almost complete degradation occurs within 25 min of irradiation time. About 54.46% of the dye was removed within 5 min, whereas the MB removal efficiency was 94.73% for 15 min of irradiation. After that, up to 25 min, dye removal efficiency increased slightly as the available dye molecules remaining in the solution were reduced [50].

### 2.15. Role of Radical Scavengers

To understand the reactive species involved in the B–ZnO/TiO$_2$ photocatalytic degradation process, scavenger studies were carried out under optimized conditions. Three different chemical scavengers, namely ascorbic acid, 2-propanol, and ammonium oxalate were employed to understand the roles of superoxide radical ions ($\bullet O_2^-$), hydroxyl radicals ($\bullet OH$), and photogenerated holes (h$^+$), respectively [7,50,51]. The photocatalyst was added to the MB solution after the scavengers. The removal efficiency of MB obtained with radical scavengers and B–ZnO/TiO$_2$ photocatalyst is illustrated in Figure 13. With the introduction of the various scavengers, it was discovered that MB dye removal efficiency was decreased to varying extents. Without any scavenging agent, higher removal efficiency of 94.73% is found for the B–ZnO/TiO$_2$ photocatalyst. On the other hand, with the addition of ascorbic acid, 2-propanol, and ammonium oxalate, dye removal efficiency is reduced to 61.27%, 60.26%, and 85.29%, respectively. The inhibition of active radicals by various scavengers has led to a significant reduction in the degradation of MB dye [7]. Even though all the radicals were involved in the degradation of the MB dye, the important roles in the photocatalytic

degradation of MB were played by the •OH and •$O_2^-$ radicals rather than the h$^+$ radical under sunlight.

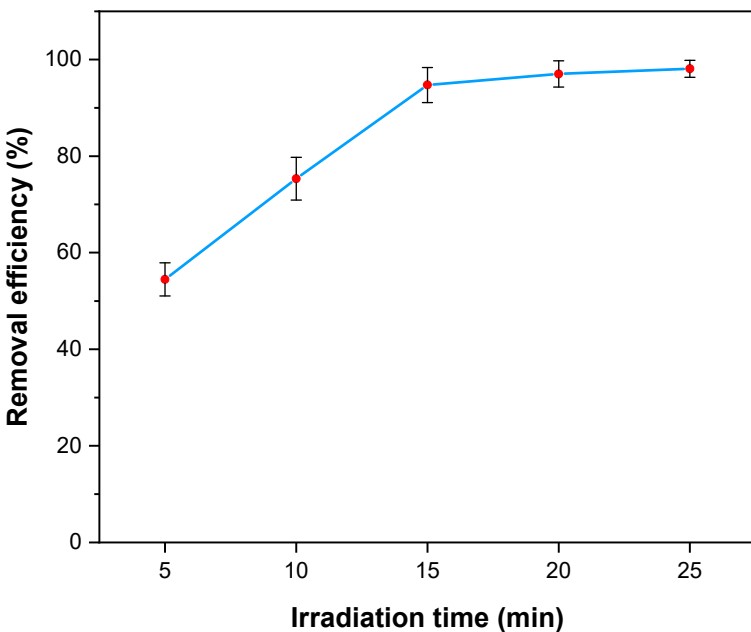

**Figure 12.** Effect of time on the photocatalytic removal of MB with B–ZnO/TiO$_2$ (initial MB concentration: 15 mg/L; photocatalyst amount: 30 mg; pH: 11; irradiation time: 5–25 min).

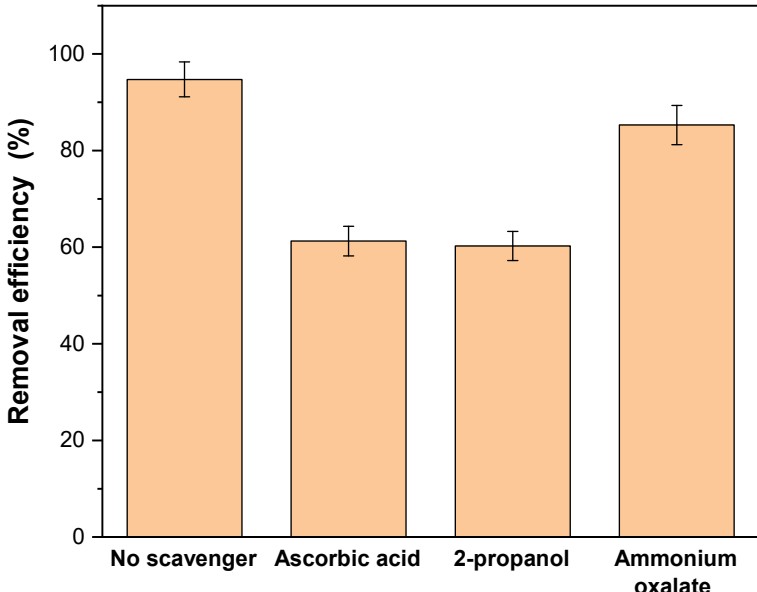

**Figure 13.** Role of radical scavengers in the photocatalytic removal of MB with B–ZnO/TiO$_2$ (initial MB concentration: 15 mg/L; concentration of all scavengers: 1 mmol/L; photocatalyst amount: 30 mg; pH: 11; irradiation time: 15 min).

### 2.16. Photocatalytic Degradation Mechanism

The following Equations (3) and (4) were used to compute the conduction band (CB) and valence band (VB) of ZnO and TiO$_2$ [35].

$$E_{CB} = \chi - E_C - 0.5E_g \tag{3}$$

$$E_{VB} = \chi - E_C + 0.5E_g \tag{4}$$

where $E_{CB}$ and $E_{VB}$ are the CB and VB edge potentials, respectively, and $E_g$ is the bandgap energy of ZnO ($\sim$2.89 eV) and TiO$_2$ ($\sim$3.06 eV). $\chi$ represents the electronegativity of ZnO (5.79 eV) and TiO$_2$ (5.89 eV), while $E_C$ is the energy of free electron on the normal hydrogen electrode scale (4.5 eV) [52,53]. The calculated CB and VB band edge potentials for ZnO are $-0.16$ and 2.73 eV, respectively, while the edge potentials for TiO$_2$ are $-0.14$ and 2.92 eV. Figure 14 schematically depicts the photocatalytic breakdown of MB by B–ZnO/TiO$_2$ nanocomposites. Electrons (e$^-$) and holes (h$^+$) are created from the VB to the CB of ZnO and TiO$_2$, respectively, when exposed to sunlight. The ECB value of ZnO ($-0.16$ eV) is more negative than that of TiO$_2$ ($-0.14$ eV). As a result, a minor amount of e$^-$ migrates from the CB of ZnO to the CB of TiO$_2$, while h$^+$ moves from the VB of TiO$_2$ to the VB of ZnO, enhancing charge separation [54]. The oxygen (O$_2$) and water (H$_2$O) in MB aqueous media can be changed to superoxide ($\bullet$O$_2^-$) and hydroxyl ($\bullet$OH) radicals by e$^-$ and h$^+$, respectively [55]. Scavenger studies also confirm that the $\bullet$O$_2^-$ and $\bullet$OH radicals are the most active species for photocatalytic degradation of MB molecules. Due to high oxidation properties, $\bullet$O$_2^-$ and $\bullet$OH radicals can react with MB dye and produce CO$_2$, H$_2$O, and other nontoxic products [23]. The MB photocatalytic mechanism of B–ZnO/TiO$_2$ nanocomposites is expressed as follows in Equations (6)–(11) [33,50,56]:

$$\text{B-ZnO/TiO}_2 + \text{Sunlight } (h\nu) \rightarrow \text{ZnO/TiO}_2(e^-) + \text{ZnO/TiO}_2(h^+) \tag{5}$$

$$\text{ZnO/TiO}_2(e^-) + \text{O}_2 \rightarrow \text{ZnO/TiO}_2 + \bullet\text{O}_2^- \tag{6}$$

$$\text{ZnO/TiO}_2(h^+) + \text{H}_2\text{O} \rightarrow \text{ZnO/TiO}_2 + \bullet\text{OH} + \text{H}^+ \tag{7}$$

$$\text{ZnO/TiO}_2(e^-) + \bullet\text{O}_2^- + 2\text{H}^+ \rightarrow \text{ZnO/TiO}_2 + \text{H}_2\text{O}_2 \tag{8}$$

$$\text{ZnO/TiO}_2(e^-) + \text{H}_2\text{O}_2 \rightarrow \text{ZnO/TiO}_2 + \bullet\text{OH} + \text{OH}^- \tag{9}$$

$$\text{MB dye} + \bullet\text{O}_2^- / \bullet\text{OH} \rightarrow \text{Intermediates} \rightarrow \text{CO}_2 + \text{H}_2\text{O} \tag{10}$$

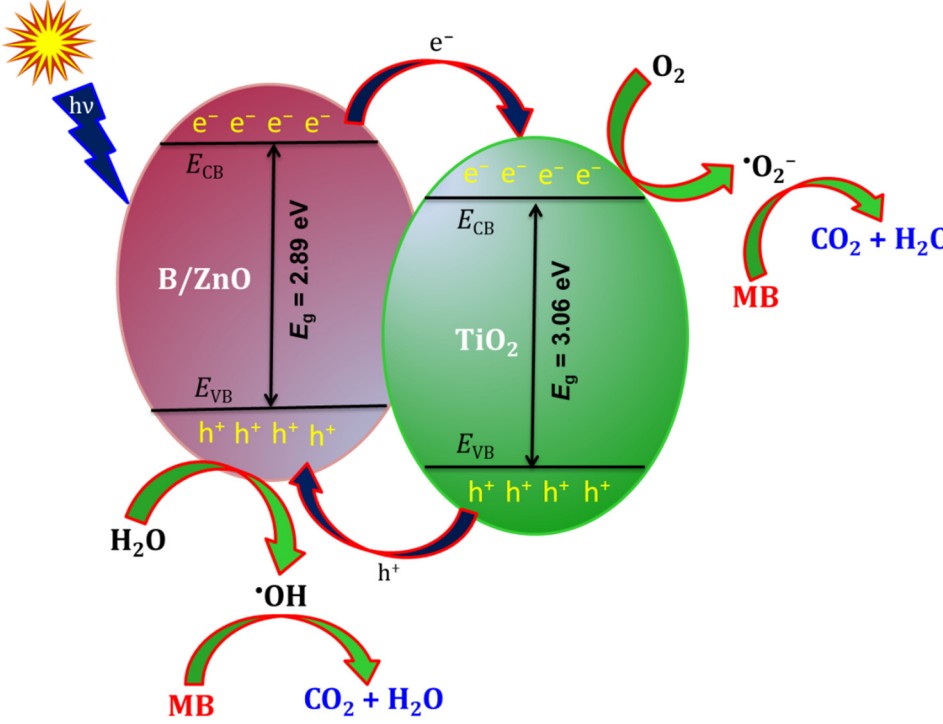

**Figure 14.** Schematic diagram of B–ZnO/TiO$_2$ photocatalytic degradation mechanism.

## 2.17. Photocatalyst Reusability

The reusability of the used B–ZnO/TiO$_2$ photocatalyst was investigated up to five cycles at the optimal conditions (initial MB concentration: 15 mg/L; photocatalyst amount:

30 mg; pH: 11; irradiation time: 15 min), and the result of the reusability experiments is illustrated in Figure 15. For this purpose, the used photocatalyst was collected and dried for the next cycle [35]. It was found that the MB removal efficiency was marginally decreased for the B–ZnO/TiO$_2$ photocatalyst in the successive cycles. Because of the poisoning effect of degradation products and the blockage of sunlight irradiation, the dye removal efficiency gradually declined through five cycles [50]. Hence, it can be inferred that B–ZnO/TiO$_2$ is appropriate for the removal of dye from textile wastewater over a long time and with repeated application.

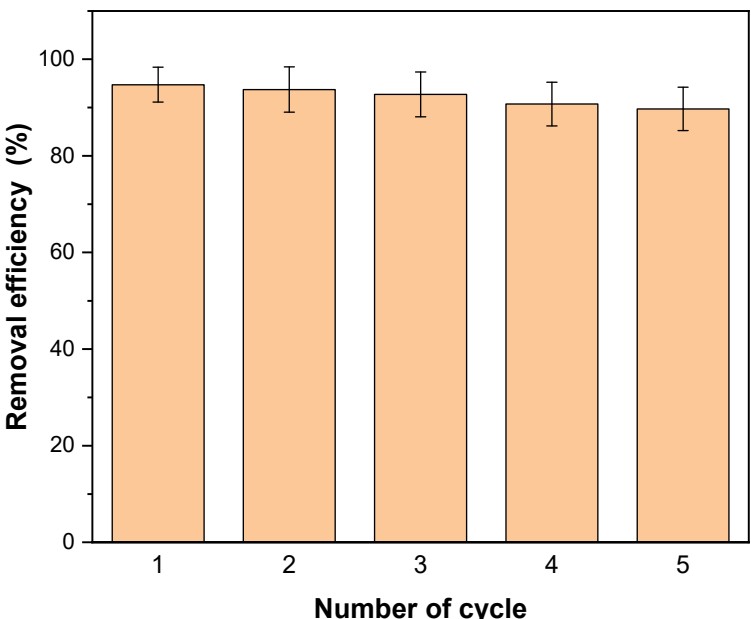

**Figure 15.** The cycles of photocatalytic experiment of B–ZnO/TiO$_2$ after five cycles (initial MB concentration: 15 mg/L; photocatalyst amount: 30 mg; pH: 11; irradiation time: 15 min).

## 3. Materials and Methods

### 3.1. Materials

Zinc acetate dihydrate (Zn(CH$_3$COO)$_2$·2H$_2$O), oxalic acid dihydrate ((COOH)$_2$·2H$_2$O), boric acid (H$_3$BO$_3$), commercial TiO$_2$, methylene blue (C$_{16}$H$_{18}$ClN$_3$S), ascorbic acid (C$_6$H$_8$O$_6$), ammonium oxalate ((NH$_4$)$_2$C$_2$O$_4$), and 2-propanol ((CH$_3$)$_2$CHOH) were obtained from Merck (Darmstadt, Germany). Purified sodium hydroxide (NaOH) pellets and hydrochloric acid (HCl, 37%) were obtained from Active Fine Chemicals Limited (Dhaka, Bangladesh). All the chemicals were analytical grade and were not purified before use. For the preparation of solutions, deionized water was used.

### 3.2. Synthesis of B–ZnO/TiO$_2$

B–ZnO/TiO$_2$ composites were synthesized using a mechanochemical–calcination method with a controlled combustion method. Figure S4 represents the synthesis process used for the fabrication of B–ZnO/TiO$_2$. For this experiment, the mixture of zinc acetate dihydrate (2.195 g) and oxalic acid dihydrate (2.521 g) was ground for 10 min in an agate mortar to form a paste of zinc oxalate and acetic acid. To obtain the precursor of B–ZnO/TiO$_2$ (5 wt% B doped with molar ratio 1:1.4 of ZnO/TiO$_2$), boric acid and commercial TiO$_2$ were introduced to the above paste, followed by grinding after each addition. The precursor was calcined for 3 h at 500 °C in an air atmosphere to prepare the B–ZnO/TiO$_2$ composites [57]. For comparison, ZnO, B–ZnO, and ZnO/TiO$_2$ were also synthesized with the use of the same conditions and without/with the respective additions of boric acid and commercial TiO$_2$.

### 3.3. Characterization

An X-ray diffractometer (Ultima IV, Rigaku Corporation, Akishima, Japan) equipped with Cu K$\alpha$ radiation ($\lambda$ = 0.154 nm) was used to evaluate the XRD patterns of the samples at room temperature. The XRD data were measured over a scanning range of 10–70° ($2\theta$). The machine was maintained at 40 mA and 40 kV. The scanning step size was 0.02° and the scanning speed was 3° min$^{-1}$. The surface morphology and particle size distribution of the samples were analyzed by an analytical field emission scanning electron microscope (FESEM, JSM 7600F, JEOL, Japan) equipped with EDS to determine the elemental composition. For FESEM measurements, the accelerating voltage was 5.0 kV, and for EDS analysis, the accelerating voltage was 10 kV. For the identification of the functional groups present in the samples, a Fourier transform infrared spectrometer (FTIR, FT-IR 8400S spectrophotometer (Shimadzu Corporation, Kyoto, Japan)) was used (in the wavenumber range of 4000–400 cm$^{-1}$; resolution: 4 cm$^{-1}$; scans: 30). The diffuse reflectance spectra (DRS) of the photocatalysts in the range of 300–700 nm were recorded with a UV-3600i Plus UV-Vis-NIR spectrophotometer (Shimadzu Corporation, Japan). The textural properties of the adsorbents were characterized by nitrogen adsorption–desorption at −196 °C after evacuation for 12 h at 110 °C using a surface area and porosity analyzer (BET Sorptometer, BET-201-A, PMI, Tampa, FL, USA). According to the Brunauer–Emmett–Teller (BET) equation, the surface areas of the photocatalysts were calculated. Photoluminescence (PL) spectra were recorded using a Shimadzu RF-6000 system (Shimadzu Corporation, Kyoto, Japan) equipped with a 150 W xenon lamp at room temperature. The excitation wavelength for each sample was 325 nm, and the emission wavelengths ranged from 200 to 800 nm. X-ray photoelectron spectroscopy (XPS) measurements were carried out with a Thermo Scientific photoelectron spectrometer using Al K$\alpha$ (monochromatic, 1486.6 eV) radiation (UK).

### 3.4. Removal of MB

The dye removal performance of commercial $TiO_2$, ZnO, B–ZnO, $ZnO/TiO_2$, and B–$ZnO/TiO_2$ photocatalysts was assessed at different operating conditions, including the pH, photocatalyst amount, and initial dye concentration, employing a standard solution of MB ($\lambda_{max}$ = 662 nm [11]) under solar irradiation in open air. Typically, 50 mL of standard MB solution was added to a 250 mL beaker along with 30 mg of synthesized photocatalyst, as illustrated in Figure S5. After exposing the suspensions in the beaker to sunlight, approximately 3 mL of MB solution was collected and filtered using an Advantec membrane filter of 0.45 μm at various time intervals. The remaining concentration of MB after degradation at various conditions was determined by recording absorbance on a UV-1700 Spectrophotometer (Shimadzu, Japan). Batch experiments were conducted under identical conditions (location coordinates: 23.7275° N, 90.4019° E; temperature: ~30 °C; time of year: October to November) on sunny days between 11:00 and 14:00 BST (Bangladesh standard time) for the determination of the percentage removal of MB using the studied photocatalysts. A UV radiometer (UVR-400, Iuchi Co., Osaka, Japan) with a 320–420 nm wavelength sensor was used to measure the intensity of light, as presented in Table S2. To modify the pH of the solutions prior to degradation experiments, aqueous solutions of dilute HCl (0.1 M) and NaOH (0.1 M) were utilized. It is noted that the natural solution pH of MB was ~6.0 [5]. In reusability studies, the catalyst was filtered after each cycle and a fresh MB solution with the same concentration was used. The following equation was used to determine the MB removal efficiency:

$$\text{MB removal efficiency } (\%) = \frac{C - C_0}{C} \times 100 \tag{11}$$

Here, $C$ and $C_0$ denote the initial and final MB concentrations. All of the experiments were conducted in triplicate, and the results were presented as mean values. The relative standard deviations ranged from 2.5% to 12%.

## 4. Conclusions

An unprecedented B–ZnO/TiO$_2$ nanophotocatalyst was synthesized by the mechano-chemical–calcination method. Because of the synergistic effects of boron, ZnO, and TiO$_2$, this catalyst exhibits superior properties and capabilities towards the completion of photo-catalytic removal of toxic textile dyes. The synthesized photocatalyst exhibits a crystallite size of 42.54 nm, a BET surface area of 18.99 m$^2$/g, and bandgap energies of 2.89 eV (direct transitions) and 3.06 eV (indirect transitions). The noteworthy reduction in bandgap energy influenced the extension of optical absorption towards the visible-light region, and signifi-cant PL quenching in the B–ZnO/TiO$_2$ photocatalyst spectrum confirmed the reduction in electron–hole recombination rate, which amends the shortcoming of TiO$_2$ and ZnO. Hence, the B–ZnO/TiO$_2$ photocatalyst demonstrates better photocatalytic activity, and the MB photocatalytic removal efficiency of ~95% is observed within 15 min of natural sunlight irradiation. The scavenger test was used to determine the probable mechanism of photocatalytic degradation of MB by B–ZnO/TiO$_2$. The photocatalyst is stable for up to five cycles. Therefore, the synthesized B–ZnO/TiO$_2$ photocatalyst could be a potential alternative for removing hazardous textile dyes.

**Supplementary Materials:** The following supporting information can be downloaded at: https://www.mdpi.com/article/10.3390/catal12030308/s1, Figure S1: Particle size distribution histograms of (a) commercial TiO$_2$, (b) ZnO, and (c) B–ZnO/TiO$_2$, Figure S2: (a) SEM image of B–ZnO/TiO$_2$ and EDS elemental mapping of (b) boron (B), (c) zinc (Zn), (d) titanium (Ti), and (e) oxygen (O), Figure S3: EDS pattern of (a) commercial TiO$_2$, (b) ZnO, (c) B–ZnO, (d) ZnO/TiO$_2$, and (e) B–ZnO/TiO$_2$, Figure S4: Schematic representation of the synthesis process of B–ZnO/TiO$_2$, Figure S5: Schematic diagram of the reactor for photocatalytic degradation of MB dye, Table S1: Elemental analysis of commercial TiO$_2$, ZnO, B–ZnO, ZnO/TiO$_2$, and B–ZnO/TiO$_2$ from EDS, Table S2: Sunlight intensity during different experiments.

**Author Contributions:** Conceptualization, M.A.I.M.; methodology, R.A.S. and M.A.I.M.; software, R.A.S. and M.S.Q.; validation, R.A.S. and M.S.Q.; formal analysis, S.A.F., M.S. and M.A.I.M.; investiga-tion, R.A.S.; resources, M.M., S.M.M. and M.A.I.M.; data curation, M.S.Q. and M.M.; writing—original draft preparation, R.A.S. and M.A.I.M.; writing—review and editing, R.A.S., S.A.F. and M.A.I.M.; visualization, S.A.F., M.S. and S.M.M.; supervision, S.M.M. and M.A.I.M.; project administration, S.M.M. and M.A.I.M.; funding acquisition, S.M.M. and M.A.I.M. All authors have read and agreed to the published version of the manuscript.

**Funding:** This research was funded by the Centennial Research Grant (CRG) from the University of Dhaka, Bangladesh (Ref. No. Reg/Admin-3/13999).

**Data Availability Statement:** Data are contained within the article and Supplementary Materials.

**Acknowledgments:** The authors are grateful to the University of Dhaka, Bangladesh for financial support. We are thankful to the Centre for Advanced Research in Sciences (CARS), University of Dhaka, Bangladesh, the Department of Physics at Bangladesh University of Engineering and Technol-ogy (BUET), Dhaka, Bangladesh, and the Bangladesh Council of Scientific and Industrial Research (BCSIR), Dhaka, Bangladesh, for providing partial analytical support to carry out this research.

**Conflicts of Interest:** The authors declare no conflict of interest. The funders had no role in the design of the study; in the collection, analyses, or interpretation of data; in the writing of the manuscript; or in the decision to publish the results.

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
