# Peer review of "Natural Sunlight Driven Photocatalytic Removal of Toxic Textile Dyes in Water Using B-Doped ZnO/TiO2 Nanocomposites"

_catalysts, doi:10.3390/catal12030308_

Round 1

Reviewer 1 Report

The manuscript entitled “Natural Sunlight Driven Photocatalytic Removal of Toxic Tex- 2

tile Dyes in Water Using B–Doped ZnO/TiO2 Nanocomposites” has been submitted by authors. Some issues to be addressed which will improve the quality of manuscript. Therefore, I recommend this work could be published after the major revision

  1. The English composition requires many improvements. The authors should proofread the manuscript carefully to minimize grammatical errors.
  2. The background of this work is not clear. The authors should specify in a clearer way what novel and original this work proposes to readers based on some new works. This research topic is widely studied in past and lot of studies are performed.
  3. The characterization part and the result and discussion part are not supported by enough references. It may be supported by the recent relevant references (before 2015).

Scientific Reports, 6, 20103 (2016); Journal of Alloys and Compounds, 886, 2021, 161169.

Author Response

Dear Reviewer-1

Thank you very much for your sincere comments. The manuscript has been greatly improved with your comments. The detailed replies to your comments are listed below, and the revised texts are marked as red.

The manuscript entitled “Natural Sunlight Driven Photocatalytic Removal of Toxic Textile Dyes in Water Using B–Doped ZnO/TiO2 Nanocomposites” has been submitted by authors. Some issues to be addressed which will improve the quality of manuscript. Therefore, I recommend this work could be published after the major revision

C1. The English composition requires many improvements. The authors should proofread the manuscript carefully to minimize grammatical errors.

Ans: Thank you for your suggestion. We checked and improved the English composition to minimize grammatical errors.

C2. The background of this work is not clear. The authors should specify in a clearer way what novel and original this work proposes to readers based on some new works. This research topic is widely studied in past and lot of studies are performed.

Ans: Thank you very much for your valuable comment. Based on your suggestion, we have strengthened the novelty and originality of this study in the concluding paragraph of the introduction. Please check.

C3. The characterization part and the result and discussion part are not supported by enough references. It may be supported by the recent relevant references (before 2015). Scientific Reports, 6, 20103 (2016); Journal of Alloys and Compounds, 886, 2021, 161169.

Ans: We highly appreciate your advice. Two references (Scientific Reports, 6, 2016, 20103; Journal of Alloys and Compounds, 886, 2021, 161169) have been cited in the result and discussion part of the manuscript (Ref. No. 26 and 32).

Reviewer 2 Report

The manuscript describes the synthesis and the photocatalytic activity of titania/zinc oxide nanocompounds doped with boron. The topic is interesting and the research activities are well described. Photocatalytic activity was measured using natural solar radiation and special care was taken to ensure reasonable reproducibility of the experimental conditions.

I recommend the publication of the manuscript on “Catalysts” after the addressing of the few minor issues.

Minor issues

1) In the experimental section it is stated that the solar irradiation during photocatalytic experiments was measured with an UV radiometer (row 518). The irradiance data are very interesting in order to complete the description of the experimental conditions and should be reported in the manuscript (ideally as the irradiance recorded for each experiment, maybe in a dedicated table in Supplementary Information).

2) Specify the exact meaning of the error bars (e.g. estimated standard deviation, confidence interval, etc.) in the captions of figures 8c, 9, 10, 11, 12, 13, 15.

3) Bar plot are suited for categorized data but not for reporting data points along a continuous variable as pH (Fig. 9), photocatalyst amount (Fig. 10), dye concentration (Fig. 11), irradiation time (Fig. 12). For these kind of data sets is more suited a scatter plot (such as in Fig. 5). The indicated bar plots should therefore be converted in scatter plots. Note that the bar plot in Fig. 15 is correct because the number of cycle is not a continuous variable.

4) Use colors sparingly in plots and only if necessary in order to identify different data sets. The colorization of bars in Fig. 13 and 15 is useless and should be avoided (this applies also to the other bar plots in Fig. 9, 10, 11 and 12 that should be converted anyway in scatter plot)

Author Response

Dear Reviewer-2

Thank you very much for your valuable comments. The manuscript has been greatly improved with your suggestions. The detailed replies to your comments are listed below, and the revised texts are marked as red.

The manuscript describes the synthesis and the photocatalytic activity of titania/zinc oxide nanocompounds doped with boron. The topic is interesting and the research activities are well described. Photocatalytic activity was measured using natural solar radiation and special care was taken to ensure reasonable reproducibility of the experimental conditions. I recommend the publication of the manuscript on “Catalysts” after the addressing of the few minor issues.

C1. In the experimental section it is stated that the solar irradiation during photocatalytic experiments was measured with an UV radiometer (row 518). The irradiance data are very interesting in order to complete the description of the experimental conditions and should be reported in the manuscript (ideally as the irradiance recorded for each experiment, maybe in a dedicated table in Supplementary Information).

Ans: Thank you very much for your valuable comments. The irradiance data of different photocatalytic experiments have been presented in Table S2 in Supplementary Information.

C2. Specify the exact meaning of the error bars (e.g. estimated standard deviation, confidence interval, etc.) in the captions of figures 8c, 9, 10, 11, 12, 13, 15.

Ans: Thank you for your query. The following information has been added in experimental section as: “All of the experiments were done in triplicate, and the results were presented as mean values. The relative standard deviations ranged from 2.5–12%”.

C3. Bar plot are suited for categorized data but not for reporting data points along a continuous variable as pH (Fig. 9), photocatalyst amount (Fig. 10), dye concentration (Fig. 11), irradiation time (Fig. 12). For these kind of data sets is more suited a scatter plot (such as in Fig. 5). The indicated bar plots should therefore be converted in scatter plots. Note that the bar plot in Fig. 15 is correct because the number of cycle is not a continuous variable.

Ans: Your suggestions are greatly appreciated. Figures 9, 10, 11, and 12 have been changed from bar plots to scatter plots.

C4. Use colors sparingly in plots and only if necessary in order to identify different data sets. The colorization of bars in Fig. 13 and 15 is useless and should be avoided (this applies also to the other bar plots in Fig. 9, 10, 11 and 12 that should be converted anyway in scatter plot).

Ans: Thank you for your advice. The colorization of bars in Figures 13 and 15 has been changed accordingly.

Reviewer 3 Report

This work proposed a B-doped ZnO/TiO2 nanocomposite photocatalyst to remove methylene blue (MB) dye under natural sunlight irradiation. Boron doping can effectively increase the surface area and reduce the band gap energy of ZnO/TiO2. The MB removal efficiency can reach to ~95% under the optimized condition. The high stability and reusability of B-doped ZnO/TiO2 shows it is an economical and environmentally friendly photocatalyst. Several comments are required to be addressed prior to the acceptance.

  1. Please clarify the B doping concentration and mass/molar ratio of ZnO to TiO2 in B-doped ZnO/TiO2
  2. By incorporating ZnO into TiO2, FESEM image in Figure 2d shows a higher degree of agglomeration. The ultrasonic treatment in proper solvent may improve the dispersion of small particles. The SEM, BET, and photocatalyst property are comparable if all samples are treated in ultrasonic dispersion.
  3. The working mechanism of B on ZnO/TiO2 morphology should be clarified, especially on the particle size. Without B doping, ZnO/TiO2 shows small particle size in Figure 2d. However, B doped ZnO/TiO2 shows much larger particle size in Figure 2e.
  4. The scale bar in Figure 2 is hard to read.
  5. B-O stretching vibrations were observed in FTIR study. There is no evidence to show that B was doped into ZnO/TiO2 EDS mapping or SEM image in back-scatter mode is recommended to show the element distribution.
  6. XPS spectra in Figure 7e shows the existence of C-O, C=O, and OH. If calcinated at higher temperature (e.g., 600 oC), will these groups be removed? It may increase the active catalyst mass ratio after the impurities removed/burned and hence increase the photocatalysis activity.
  7. The removal efficiency when pH is 12 is higher than that of 11. Why pH=11 was taken as the optimum condition? What is the side effect of higher pH for the photocatalyst operation?

Author Response

Dear Reviewer-3

Thank you very much for your valuable query. The manuscript has been greatly improved with your advice. The detailed replies to your comments are listed below, and the revised texts are marked as red.

This work proposed a B-doped ZnO/TiO2 nanocomposite photocatalyst to remove methylene blue (MB) dye under natural sunlight irradiation. Boron doping can effectively increase the surface area and reduce the band gap energy of ZnO/TiO2. The MB removal efficiency can reach to ~95% under the optimized condition. The high stability and reusability of B-doped ZnO/TiO2 shows it is an economical and environmentally friendly photocatalyst. Several comments are required to be addressed prior to the acceptance.

C1. Please clarify the B doping concentration and mass/molar ratio of ZnO to TiO2 in B-doped ZnO/TiO2.

Ans: Thank you for your query. The B doping concentration and molar ratio of ZnO to TiO2 have been added in the Synthesis section as: 5 wt% B doped with molar ratio 1:1.4 of ZnO/TiO2.

C2. By incorporating ZnO into TiO2, FESEM image in Figure 2d shows a higher degree of agglomeration. The ultrasonic treatment in proper solvent may improve the dispersion of small particles. The SEM, BET, and photocatalyst property are comparable if all samples are treated in ultrasonic dispersion.

Ans: Thank you very much for your valuable comments. Following that, we will take your advice.

C3. The working mechanism of B on ZnO/TiO2 morphology should be clarified, especially on the particle size. Without B doping, ZnO/TiO2 shows small particle size in Figure 2d. However, B doped ZnO/TiO2 shows much larger particle size in Figure 2e.

Ans: Thank you for your query. The working mechanism of B on ZnO/TiO2 morphology has been added in FESEM Study section as:

Generally, B doping deteriorates the crystallinity of ZnO to a certain extent [1], because the radius of B3+ (0.02 nm) is smaller than that of O2− (0.14 nm) and Zn2+ (0.074 nm). As a result, when the O or Zn atoms are replaced with B, the crystal plane spacing shrinks, causing the diffraction peaks to shift to a greater angle. Consequently, the B atoms are likely to occupy the octahedral interstices [1]. However, Figure 2e exhibited substantially larger particles because the FESEM image of the heterogeneous semiconductor photocatalyst B–ZnO/TiO2 was focused where TiO2 nanoparticles were more predominant. To make it easier to understand, another FESEM picture of B–ZnO/TiO2 nanocomposites (Figure 2f) has been presented, which shows characteristics of both TiO2 and B–ZnO.  

C4. The scale bar in Figure 2 is hard to read.

Ans: 100 nm scale bar has been added in each figures of FESEM.

C5. B-O stretching vibrations were observed in FTIR study. There is no evidence to show that B was doped into ZnO/TiO2 EDS mapping or SEM image in back-scatter mode is recommended to show the element distribution.

Ans: Thank you for your query. However, the Boron peak is visible in the EDS pattern of B-ZnO/TiO2 (Figure S3e). Boron is also visible in the XPS spectra of B–ZnO/TiO2 (Figure 7a and Figure 7d). Moreover, based on your suggestion, SEM and EDS mapping examinations were carried out on B–ZnO/TiO2 (Figure S2). It observed that the various elements of B, Zn, Ti, and O were distributed throughout the sample, confirming the presence of B in B–ZnO/TiO2 nanocomposites.

C6. XPS spectra in Figure 7e shows the existence of C-O, C=O, and OH. If calcinated at higher temperature (e.g., 600 oC), will these groups be removed? It may increase the active catalyst mass ratio after the impurities removed/burned and hence increase the photocatalysis activity.

Ans: Thank you very much for your valuable suggestion. Following that, we will take your advice. However, calcination at 600 oC will convert TiO2 from the anatase phase to the rutile phase. It is considered that anatase TiO2 is more effective for photocatalysis than the rutile phase.

C7. The removal efficiency when pH is 12 is higher than that of 11. Why pH=11 was taken as the optimum condition? What is the side effect of higher pH for the photocatalyst operation?

Ans: Thanks for your valuable comments. According to Table 6.7.2 of the literature [doi:10.1016/B978-0-12-810391-3.00006-0], pH of textile waste effluent is around 10. Also removal efficiency above 90% is considered to be sufficient. At pH 11, the removal efficiency is ∼95% whereas at pH 12, the removal efficiency is ∼98. There is not much difference in removal efficiency between pH 11 and pH 12. Furthermore, the highly basic pH 12 solution will cause another source of pollution. 

Round 2

Reviewer 1 Report

Author address all comments very carefully,  its ready to accept  in present form.